# Antimicrobial and Antibiofilm Potential of *Thymus vulgaris* and *Cymbopogon flexuosus* Essential Oils against Pure and Mixed Cultures of Foodborne Bacteria

**DOI:** 10.3390/antibiotics12030565

**Published:** 2023-03-13

**Authors:** Joana Monteiro Marques, Susana Serrano, Hiba Selmi, Pedro Giesteira Cotovio, Teresa Semedo-Lemsaddek

**Affiliations:** 1Centre for Interdisciplinary Research in Animal Health (CIISA), Faculty of Veterinary Medicine, University of Lisbon, 1649-004 Lisboa, Portugal; 2Associate Laboratory for Animal and Veterinary Sciences (AL4AnimalS), Portugal; 3Laboratory of Protein Engineering and Bioactive Molecules (LIP-MB), National Institute of Applied Sciences and Technology, University of Carthage, Carthage 1054, Tunisia; 4Faculty of Sciences of Bizerta, University of Carthage, Carthage 1054, Tunisia; 5LASIGE, Faculty of Sciences, University of Lisbon, 1649-004 Lisboa, Portugal

**Keywords:** foodborne, pathogenic, spoilage bacteria, biofilm, essential oils, antimicrobial, anti-biofilm, in silico absorption and toxicity prediction

## Abstract

The spread of pathogenic and food spoilage microorganisms through the food chain still faces major mitigation challenges, despite modern advances. Although multiple cleaning and disinfection procedures are available for microbial load reduction in food-related settings, microbes can still remain on surfaces, equipment, or machinery, especially if they have the ability to form biofilms. The present study assessed the biofilm-forming properties of pure and mixed cultures of foodborne and spoilage bacteria (*Listeria monocytogenes*, *Enterococcus faecalis*, *Aeromonas hydrophila*, *Brochothrix thermosphacta*), using polystyrene and stainless steel contact surfaces. Subsequently, the antimicrobial and antibiofilm properties of *Thymus vulgaris* and *Cymbopogon flexuosus* essential oils—EOs—were evaluated against these bacteria. Moreover, in silico prediction of the absorption and toxicity values of the EOs’ major constituents was also performed, perceiving the putative application in food-related settings. Overall, biofilm formation was observed for all microbes under study, at different temperatures and both contact surfaces. In polystyrene, at 25 °C, when comparing pure with mixed cultures, the combination *Listeria–Aeromonas* achieved the highest biofilm biomass. Moreover, at 4 °C, increased biofilm formation was detected in stainless steel. Regarding thyme, this EO showed promising antimicrobial features (especially against *A. hydrophila,* with a MIC of 0.60 µg/µL) and antibiofilm abilities (MBEC of 110.79 µg/µL against *L. monocytogenes,* a major concern in food settings). As for lemongrass EO, the highest antimicrobial activity, with a MIC of 0.49 µg/µL, was also observed against *L. monocytogenes*. Overall, despite promising results, the in situ effectiveness of these essential oils, alone or in combination with other antimicrobial compounds, should be further explored.

## 1. Introduction

Food spoilage and foodborne diseases continue to occur worldwide, which has raised awareness regarding the urgent need to ensure food quality and safety. Briefly, food spoilage is considered an alteration of a food’s quality, which makes it undesirable and/or unsuitable for consumption, due to odor, appearance or texture alterations [1]. Moreover, the consumption of contaminated food products results in approximately 600 million cases of foodborne illnesses and 420,000 deaths per year [2]. In the food industry, spoilage-inducing and pathogenic microbes may be present, either in a planktonic or biofilm state [3]. Although several cleaning and disinfection procedures are commonly applied in order to reduce microbial load, bacteria can remain on food, food contact surfaces, equipment, or machinery [4], especially if they have the ability to produce biofilms. A biofilm is a community of sessile bacteria surrounded by an extracellular polymeric substance (EPS), which confers resistance to environmental changes and antimicrobial agents, allowing and improving their ability to colonize surfaces [5]. In food industry settings, biofilms can be formed on mixing tanks, vats, tubing, or other stainless steel structures, constituting a serious problem, which may lead to economic losses and/or a higher risk for cross-contamination and foodborne diseases [6]. *Listeria monocytogenes*, *Staphylococcus aureus*, *Salmonella* spp., *Escherichia coli*, *Bacillus cereus*, *Enterococcus* spp., and *Campylobacter jejuni*, among many other microorganisms, have been documented as having the capacity to form biofilms on food and food contact surfaces [3] in either pure forms or as mixed microbial communities [7]. Moreover, there are various research papers studying residential bacteria of food industry locations that have the ability to produce biofilms, in order to assess their potential to either promote or inhibit the growth of foodborne pathogenic bacteria (*L. monocytogenes*, for instance), as well as investigate their use as a contamination control method [8,9]. Hence, to face this problem, several methodologies have been developed and applied over the years [10,11,12]. However, the generalized use of chemical biocides may be ineffective, due to biofilm protection effects and the development of resistance mechanisms [13]. Therefore, in recent years, natural antimicrobials have emerged as eco-friendly alternatives—green biocides—among which are the plant-derived essential oils (EOs) [5]. EOs are plant extracts, consisting of a variety of aromatic and volatile compounds, usually comprising between 20–60 components, at distinct concentrations. EO composition varies from plant to plant, due to differences in the plant parts used for the extraction of the bioactive compound, the extraction method, or the growth environment. EOs with associated antimicrobial properties include lavender, thyme, cinnamon, vanilla, oregano, basil, and rosemary [14,15]. Moreover, considering the particularities of the food industry, it is fundamental to highlight that several EOs harbor Generally Recognized as Safe (GRAS) status by the FDA (Food and Drug Administration, Silver Spring, MD, USA), including basil, cinnamon, clove, coriander, ginger, lavender, menthol, nutmeg, lemongrass, oregano, rose, sage, and thyme [14,16]. In recent years, *Thymus vulgaris* became one of the most studied species, as it contains bioactive components that have shown antagonistic effects against foodborne bacteria, such as *Listeria monocytogenes* [17]. Analogous results have been reported regarding thyme EO’s ability to inhibit *E. faecalis* biofilm formation, by affecting cell adherence and EPS synthesis [18]. Regarding lemongrass EO, numerous antimicrobial and food preservative applications have already been reported [19,20].

Overall, the aims of the present study were to assess the biofilm formation ability of food-related Gram-positive and Gram-negative bacteria, and to investigate the antimicrobial and antibiofilm properties of *Thymus vulgaris* and *Cymbopogon flexuosus* EOs against those foodborne pathogens and spoilage microbes in a biofilm state, using both pure and mixed cultures. Moreover, the absorption and toxicity values of the EO constituents were also in silico predicted, to investigate their potential applications in the microbial control at food-related settings.

## 2. Materials and Methods

### 2.1. Bacterial Strains and Growth Conditions

For the present study, both Gram-positive and Gram-negative food-related bacteria, including foodborne pathogens and spoilage microbes, were selected. Namely, *Enterococcus faecalis* QSE123 [21], *E. faecalis* V583 [22], *Listeria monocytogenes* CECT 937 (serotype 3b) [23], *L. monocytogenes* CECT 935 (serotype 4b) [24], *Aeromonas hydrophila* A259 (from our laboratory collection), and *Brochothrix thermosphacta* ATCC 11509^T^ [25] were selected. To facilitate the visualization of the results, the bacterial strains were given codes, which are used throughout this document (see Table 1). For all the assays that are subsequently explained, bacterial cultures were prepared in 5 mL of Brain Heart Infusion broth (BHI, Scharlau, Sentmenat, Spain) and incubated overnight at their designated optimal temperatures (Table 1). After growth, the bacterial cultures were centrifuged for 5 min at 15,885 g (in an Eppendorf centrifuge 5415R), and the microbial suspensions to be used as inoculums were prepared using BHI broth, containing approximately 10^9^ CFU/mL [26].

### 2.2. Essential Oils

For the essential oils—EOs—used in the present study, their common name, part of the plant used for extraction, origin, and major components are displayed in Table 2. To obtain EO work suspensions, each EO was diluted in 0.15% agar (Agar-Agar, Scharlau, Sentmenat, Spain) in a 1:1 (*v*/*v*) proportion, resulting in an initial concentration (work stock) of 497.6 µg/µL for thyme EO (henceforth referred to as EOT) and 499.2 µg/µL for lemongrass EO (hereafter referred to as EOL) [27]. Regarding the various EOs’ concentrations used in the assays, dilutions of 1:2 (*v*/*v*) in 0.15% agar were prepared as needed, and were always kept at −20 °C.

### 2.3. Biofilm Formation

The present study evaluated bacterial biofilm-forming capacity using polystyrene 96-well microtiter plates (Nunc ™, Thermo Fisher Scientific Inc., Roskilde, Denmark), with peg lids (Nunc ™ Transferable Solid Phases Screening System, Fisher Scientific Inc., Roskilde, Denmark), and hand-made stainless steel disks (11 mm diameter × 2 mm height). The assays were performed for both pure and polymicrobial bacterial suspensions, at two distinct temperatures, 4 and 25 °C.

#### 2.3.1. Biofilm Formation in Polystyrene

The biofilm formation assay was based on the protocol by Joe Harrison et al. (2010), where the use of lids with pegs favored biofilm production in the pegs, allowing for their simplified use for subsequent antimicrobial susceptibility assays [28]. Pure bacterial suspensions were prepared to a final concentration of 1 × 10^9^ CFU/mL, as mentioned above in Section 2.1. and inoculated to a 1:100 final dilution per well, with the following incubation conditions: 24, 48, and 72 h at 25 °C, and 5, 12, and 16 days at 4 °C. Each assay was performed in quadruplicate (technical replicates) and repeated three times (biological replicates) per strain (details in Figure 1A). In parallel, equivalent assays were performed using mixed bacterial cultures, prepared to a final volume of 100 µL (Figure 1B). Equal parts of each bacterial suspension (approximately 10^9^ CFU/mL) were used to obtain the mixtures that were tested.

After the incubation periods, each lid was washed three times with 0.1 M phosphate-buffered saline (PBS) at pH = 7.4; afterwards, the lid was placed in a new microtiter plate containing 200 µL of 0.1 M PBS, with 0.1% Tween 80 (*v*/*v*) (Sigma-Aldrich, Saint Louis, MO, USA) in each well, and was then placed in an ultrasonic bath (Grant, Ultrasonic Baths, Cardiff, UK) for 20 min. After the lid with the pegs was replaced by a conventional one, the OD was measured at 600 nm (SUNRISE, Serial number 1708003498, XFLUOR4 Version V 4.51), and the average and standard deviations of the readings were calculated, following the subtraction of the values obtained for the sterile, control well (named negative control).

#### 2.3.2. Biofilm Formation in Stainless Steel Disks

To evaluate the biofilm formation ability of the bacteria under study on stainless steel (pure and mixed cultures), the assay was performed using 24-well microtiter plates (VWR, Tissue Culture Plates, Beijing, China) and stainless steel disks (11 mm × 2 mm) under the following conditions: 4 °C for 12 days, or 25 °C for 48 h. The bacterial suspensions were prepared to a final concentration of 1 × 10^9^ CFU/mL, as described in Section 2.1, and inoculated with a 1:100 final dilution per well. Each assay was performed in triplicate, using three biological replicates per culture. After the incubation period, each disk was washed thrice with 0.1 M PBS and was placed in a new microtiter plate, containing 0.1 M PBS with 0.1% Tween 80 (*v*/*v*) (Sigma-Aldrich, Saint Louis, MO, USA). After placement of the microtiter plate in an ultrasonic bath (Grant, Ultrasonic Baths, Cardiff, UK) for 40 min, 200 µL of the suspension was retrieved from each well, and the OD was measured at 600 nm (SUNRISE, Serial number 1708003498, XFLUOR4 Version V 4.51). The average and standard deviations of the readings were calculated, following the subtraction of the values obtained for the sterile control wells (named negative control). Plate organization is displayed in Figure 2.

### 2.4. Antimicrobial Activity 

#### 2.4.1. Initial Screening: Agar Diffusion Method

The diffusion in solid medium protocol was adapted from the CLSI (2018) methodology for “dilution antimicrobial susceptibility tests for bacteria that grow aerobically” since, to our knowledge, no standard protocol has been described for testing EOs as antimicrobials [29]. Bacterial suspensions of the bacteria under study were prepared as previously described in Section 2.1, and inoculated by spreading them onto BHI (Scharlau, Sentmenat, Spain) plates. In parallel, the EOs were prepared to a final concentration of 15 µg/µL, and 10 µL droplets were carefully placed onto the surface of the inoculated BHI agar plates. After 24 h incubation at optimal growth conditions, the inhibition halos were observed and measured. Each assay was performed in triplicate.

#### 2.4.2. Determination of the Minimum Inhibitory Concentration (MIC) and Minimum Bactericidal Concentration (MBC)

The MIC assay was adapted from the previously described in CLSI (2018), using polystyrene 96-well microtiter plates, as well as sequential 1:2 dilutions of the antimicrobials under analysis. The EO suspensions were prepared at a concentration of 31.1 µg/µL for thyme and 31.2 µg/µL for lemongrass. All bacterial suspensions were prepared as described in the previous sections. After inoculation, the plates were incubated for 48 h at 25 °C. Inoculated wells, in the absence of antimicrobials, were considered positive controls (growth control), while negative controls corresponded to non-inoculated wells, containing BHI broth. All bacteria were tested in duplicate per microtiter plate (technical replicates). The Minimum Inhibitory Concentration (MIC) was defined as the lowest concentration at which no bacterial growth was observed in the microplate. After the incubation period, in order to assess the bactericidal and/or bacteriostatic activity of the EOs, 10 µL was retrieved from each well with non-visible growth, applied as droplets in BHI plates and incubated overnight at an optimal temperature. The Minimum Bactericidal Concentration (MBC) was defined as the lowest concentration for which no bacterial growth was observed on the agar plates. This assay was performed using both pure and mixed cultures. The MBC/MIC ratio was determined by dividing the MBC by the MIC values, for each strain/combination. The antibacterial effect was categorized as being either bactericidal or bacteriostatic according to the MBC/MIC ratio: when between 1 and 2, the effect was considered bactericidal, and values between 4 and 16 were deemed to be bacteriostatic [30]. 

#### 2.4.3. Determination of the Minimum Biofilm Eradication Concentration (MBEC)

The biofilm eradication assay was based on the Calgary biofilm device methodology described by Ceri et al. (1999) [31]. Briefly, the formation of biofilms by the bacteria under study was performed as described in Section 2.3.1., using polystyrene 96-well microtiter plates with peg lids. After inoculation, the plates were incubated for 48 h at 25 °C, and afterwards, the peg lid was washed thrice using 0.1 M PBS and placed in a new polystyrene 96-well microtiter plate, prepared with 1:2 dilutions of the EOs (Figure 3) for 30 min. The EOs concentrations used for this assay were selected with the knowledge that pre-formed biofilms can withstand 100–1000 times higher concentrations of antimicrobials when compared with planktonic cells [32]; therefore, the initial concentration of EOs was 100–1000 times higher than the MBC values derived from the assays detailed in Section 2.4.2. After the 30 min contact period, the lid was washed three times with 0.1 M PBS, placed in a new plate with 0.1 M PBS and 0.1% Tween 80 (*v*/*v*), and submitted to ultrasonic bath (Grant, Ultrasonic Baths, Cardiff, UK) for 20 min. Afterwards, 200 µL was retrieved from each well and the OD_600_ was measured (SUNRISE, Serial number 1708003498, XFLUOR4 Version V 4.51). Finally, 10 µL of each bacterial suspension was inoculated, as droplets, onto agar plates, and the plates were then incubated for 24 h at an optimal temperature. This last step was used to determine the Minimum Biofilm Eradication Concentration (MBEC) for each antimicrobial. This assay was performed in triplicate for both pure and mixed cultures, as previously described.

### 2.5. Data Analysis

#### 2.5.1. Statistical Analysis

All assays were replicated at least thrice. Statistical analysis was carried out using R software (version 4.2.2). Mean differences between the variables of the biofilm formation assay were analyzed with one-way ANOVA tests, and pairwise comparisons between microorganisms were performed using the estimated marginal means. Comparisons were performed as follows: for each combination of “contact surface/temperature conditions”, the biofilm production associated with the mixed cultures was compared with the corresponding pure cultures. For example, “L1 + E1, stainless steel at 25 °C” was compared with “L1, stainless steel at 25 °C” and with “E1, stainless steel at 25 °C” (independent comparisons). The normality assumption was verified by analyzing the ANOVA model residuals, with a quantile per quantile plot and a Shapiro–Wilk test of normality. The homogeneity of variance assumption was verified with the Levene test. The results are presented as mean ± standard deviation. *p*-values of <0.05 were considered to be significant (α significance level set at 5%). 

#### 2.5.2. In Silico Absorption and Toxicity Prediction

*In silico* absorption, distribution, metabolism, excretion, and toxicity (ADMET) prediction for every major constituent of the thyme and lemongrass EOs was performed, using the small-molecule pharmacokinetic prediction computational model, pkCSM, developed by Pires et al. (2015) [33]. SMILES strings of each EO constituents were retrieved from PubChem to perform the analysis.

## 3. Results and Discussion

### 3.1. Biofilm Formation by Pure and Mixed Cultures on Different Surfaces—Polystyrene and Stainless Steel

In the food industry, one of the main concerns for both producers and consumers is the safety of the food product. Hence, technologies such as pasteurization or sterilization, as well as hygiene protocols, are put in place to reduce health risks that are associated with colonization by pathogenic and/or spoilage bacteria [34]. However, these approaches rarely have effects against biofilm-producing microbes, which end up persisting in the industry settings. These resident bacteria may be found not only on the food itself, but also on diverse surfaces, resulting in possible cross-contamination sources throughout food processing [4], and eventually leading to foodborne illnesses if not properly contained [35,36,37]. 

In the present research, distinct foodborne bacteria were selected, namely, *Brochothrix thermosphacta*, *Enterococcus faecalis, Listeria monocytogenes,* and *Aeromonas hydrophila.* Previous studies had already identified the aforementioned microorganisms in the same settings, as part of mixed-species biofilms, either in food-related [9,38,39] or infection-associated [40,41] environments. Furthermore, distinct contact surfaces (polystyrene and stainless steel) and conditions (pure and mixed cultures, 4 and 25 °C, at different incubation periods) were tested. Data derived from preliminary assays, regarding biofilm production over-time, is displayed as Appendix A. These results led to the selection of the incubation conditions which resulted in maximum biofilm biomass, namely 4 °C for 12 days or 25 °C for 48 h, for both contact surfaces. Hence, all subsequent assays were performed applying the aforementioned conditions, and the corresponding results regarding biofilm production are plotted in Figure 4. For the mixed cultures, *Listeria* (L1) and *Enterococcus* (E1) were selected as being genus-representative isolates, to be used throughout the assays.

When analyzing the individual aptitude of the bacteria under study to produce biofilms on different contact surfaces at 25 °C, we verified that all the studied bacteria harbor that ability (Figure 4A). At this temperature, biofilm development in the polystyrene was very efficient for all individual microorganisms, with strains L1, E2, and A1 showing the highest production values under these conditions (Figure 4A). Concerning mixed species, we observed that the combinations of L1 + A1 and E1 + A1 revealed higher biofilm-forming abilities when compared with the remaining bacterial combinations (Figure 4B). Moreover, the evaluation of the biofilm biomass that was associated with pure and mixed cultures revealed significant differences between the mixture of L1 + A1 and L1 in pure culture (*p*-value = 0.003; Table 3) in polystyrene at 25 °C; while the same comparison of biofilms between those by the combination E1 + A1 and the corresponding individual microbes detected no significant differences (Table 3). For stainless steel, it was verified that all microorganisms had a similar ability to produce biofilms (Figure 4A), whose results were results corroborated by subsequent statistical analysis. Nonetheless, when mixed cultures were used, higher biofilm formation was detected for some of the combinations under study (Figure 4B). When a similar assay was performed, using the incubation temperature of 4 °C, it was also verified that all individual microorganisms were able to produce biofilms (Figure 4C), however, higher biomass values were observed for E2, A1, and B1, when compared to L1 and L2, which presented similar values (Figure 4C). With regards to the biofilm formation of mixed cultures at 4 °C, it was verified that in stainless steel, several comparisons (pure vs. mixed) were statistically relevant (details in Table 3). On polystyrene at 4 °C, only one combination showed statistical differences: B1 + L1 + E1 + A1 vs. A1 (Table 3). It is important to mention that previous studies had already evidenced the biofilm-forming ability of the bacterial genera under study. For example, Rosado de Castro et al. (2017) evaluated biofilm-producing *E. faecalis* and *E. faecium* isolates from cheese processing lines and tested stainless steel surfaces in a range of temperatures, from 7 to 47 °C; these authors observed biofilm production at the low temperatures [42]. As for *Aeromonas*, there have also been reports on this genus’s biofilm formation properties at different temperatures, ranging from 4 to 37 °C, as well as on different surfaces, including stainless steel [43,44]. Regarding *Brochothrix*, members of his genus are considered food spoilers, may be present throughout the food processing chain, and can produce biofilms, as previously described regarding microbial biofilms retrieved from meat processing lines [45]. Furthermore, in nature, biofilms are normally formed by microbial consortia (mixed species), composed by microorganisms working together to achieve a common goal [46]. However, this cooperation may also lead to competition within the ecological niche created, which can ultimately result in a reduction of the biofilm [47]. A work performed by Chen et al. (2019), using mixed cultures composed of *Vibrio parahaemolyticus* and *Listeria monocytogenes*, demonstrated a lower biofilm biomass when compared with pure cultures; their hypothesis is that these variations could be due to the reduction of bacterial load, due to competition, downregulation of biofilm-regulating genes, or a loss of metabolic activity in mixed cultures [47]. In our experiment, the same competition might have occurred, which could explain both the decrease in biofilm mass and the maintenance of biofilm formation that were observed in some cases, when using mixed cultures, regardless of the temperature or contact surface tested.

### 3.2. Antimicrobial Activity 

The EOs used in the present study, EOT and EOL, were selected based on previous work performed by our research team, Quendera et al. (2019) [27]. The susceptibility of the different microorganisms included in this study to the EOs was firstly screened using the agar diffusion method (qualitative), and further evaluated by the microdilution method for MIC and MBC determination (quantitative). Both methods allowed us to assess the antimicrobial efficiency of the compounds against the food-related bacteria and infer their potential for future applications in food-related settings. 

#### 3.2.1. Initial Screening: Agar Diffusion Method

The antimicrobial activity results for the agar diffusion method, with the growth inhibition zones (measured in millimeters) of each EO against the microorganisms under study, are displayed in Table 4.

No inhibitory activity of EOL was observed against the strains E1, E2, or B1, either due to the volatile composition of the EO that may have caused its evaporation, or due to uneven distribution on the media. Overall, EOT demonstrated higher inhibitory potential compared to EOL. Miladi et al. (2013) also determined the activity of EOT (*Thymus vulgaris*) by the disc diffusion method; a zone of inhibition around 41 mm was obtained for *L. monocytogenes* and one of 27.33 mm was obtained for *E. faecalis*. Recently, Kim et al. (2021) described *L. monocytogenes* growth inhibition by 57 mm with EOT (*Thymus zygis*) and by ≥85 mm with EOL (*Cymbopogon citratus*), using the agar well diffusion method [48]. The latter and other studies regarding some of the genera studied here demonstrated better inhibitory activity (>10 mm) of EO thyme and/or lemongrass when compared to our results, but a direct comparison between inhibition halos cannot be made, due to disparities in the EO source species, in the compounds present, and in the corresponding concentrations and methods used [48,49,50,51].

#### 3.2.2. Determination of Minimum Inhibitory Concentration (MIC) and Minimum Bactericidal Concentration (MBC)

##### Pure Cultures

The antimicrobial activity against pure cultures was assessed by the microdilution method in 96-well microtiter plates. Due to the hydrophobic and viscous nature of the EOs, this method has the disadvantage of requiring the addition of a surface-active agent to the stock solutions, in order to ensure contact between the EOs and the microorganisms [52]. For this purpose, the protocol was tested using 0.02% Tween 80 [52], DMSO, and ethanol to emulsify the oils, but 0.15% agar provided the best results. The MIC, MBC, and the MBC/MIC ratios, obtained by the optimized microdilution method, are represented in Table 5.

Comparing the two OEs, it was observed that EOT showed a more effective inhibitory activity than EOL, requiring lower compound concentrations (0.06–0.24 μg/μL), which is in accordance with results observed during the preliminary agar diffusion screening. Accordingly, EOs with terpene alcohols and aldehydes or phenols as their main components, such as thymol and carvacrol, have been described to harbor strong antibacterial activities [53]. Regarding EOT, equal MICs (0.24 μg/μL) were obtained for the strains of the genus *Listeria* (L1 and L2), *Enterococcus* (E1 and E2) and *Brochothrix* (B1). Regarding *Listeria*, a recent study also observed low MICs (0.5–4 μL/mL) of *T. vulgaris* EO against 12 strains of *L. monocytogenes* [17]. Moreover, the lowest MIC value was observed for A1 (*A. hydrophila*), with EOT inhibiting growth by 0.06 μg/μL. The susceptibility of *A. hydrophila* to EOT had also been observed during the agar diffusion assay. This strong antimicrobial effect of EOT (*T. vulgaris*) against *A. hydrophila* had already been demonstrated by several other authors [51,54,55]. As for EOL, MICs ranged from 0.49–7.80 μg/μL, the lowest MIC value being observed for L1 (*L. monocytogenes* CECT 937, serotype 3b) and the highest for L2 (*L. monocytogenes* CECT 935, serotype 4b). The disparities observed between these two *Listeria* strains might be due to the distinct serotypes, and consequent differences in bacterial susceptibility to lemongrass. Although Gram-positive bacteria seem to be more sensitive to EOs than Gram-negative bacteria, due to the composition of the cell wall, some EOs are able to penetrate different types of microbial cells and cause damage [56]. Accordingly, A1 (*A. hydrophila*) showed the highest susceptibility to EOT, whereas it seemed to be resistant to EOL, since no antimicrobial activity was detected against this strain. 

Regarding MBC determination, results showed that the two EOs analyzed harbor bactericidal effects against the bacteria under study. For EOT, MBC values ranged between 0.12 and 0.97 μg/μL, and for EOL, they ranged between 0.98 and 15.60 μg/mL. The lowest MBC/MIC ratio (2.00) was obtained for EOT against A1 (*A. hydrophila*), followed by L1, L2 (*L. monocytogenes* CECT 937 and 935, respectively), and E1 (*E. faecalis* QSE123), with a ratio of 2.04. Interestingly, although lower MIC values (stronger inhibition activity) were observed for EOT, EOL resulted in a lower MBC/MIC ratio in general, indicating higher bactericidal activity. 

It is important to emphasize that the main and/or minority constituents of the EOs and their concentrations, as well as the temperature, incubation time, or cell counts reported in previous publications, might have affected the observed antimicrobial activities, which compromises accurate comparisons and trends regarding the evolution of resistance [57].

##### Mixed Cultures

As microorganisms share an environment, compete for nutrients, or cooperate to survive in harsh conditions, the antimicrobial activity assays for each natural compound were repeated for mixed cultures, in the concentration range previously tested. The results for the MICs, MBCs, and MBC/MIC ratios of each EO against mixed cultures are displayed in Table 6. For this assay, only one strain of *L. monocytogenes* and *E. faecalis,* L1 and E1, respectively, were chosen for bacterial combinations.

Comparison of MIC results for the two EOs showed that EOT required lower concentrations for bacterial growth inhibition (MIC 0.12–0.97 μg/μL). In the case of EOL, MICs ranged from 0.49 to 1.95 μg/μL. As for MBCs, bactericidal quantification of EOT ranged between 0.97 and 1.94 μg/μL, while EOL attained values between 1.95 and 3.90 μg/μL.

Among the different bacterial combinations, L1 + B1 was demonstrated to be the most susceptible to both EOs, attaining the lowest MIC for each EO, namely, 0.12 μg/μL for EOT and 0.49 μg/μL for EOL. For the pure cultures of L1 and B1, the MICs were 0.24 μg/μL (see Table 5) for EOT, whereas only 0.12 μg/μL of thyme essential oil was necessary to inhibit L1 in combination with *B. thermosphacta*. A similar result was observed for EOL. However, the combination of L1 + B1 presented the highest MBC/MIC ratio for EOT (8.08) and EOL (≈ 4), indicating bacteriostatic activity. In addition, it can be pointed out that the mixed culture combining all isolates under study (L1 + E1 + A1 + B1) also showed a high MBC/MIC ratio for both EOs (≈ 4), revealing bacteriostatic activity. This might be a consequence of growth specificities in a microbial consortium, in which bacteria work for a common goal, survival [49,50].

Overall, lower MIC/MBC ratios were observed for microbial combinations, in comparison with pure cultures, for both EOs. It should be highlighted that for EOT, the combinations containing A1 (*A. hydrophila,* lower MIC for EOT), were categorized as bactericidal (ratio < 2). 

#### 3.2.3. Determination of the Minimum Biofilm Eradication Concentration (MBEC)

The aptitude of bacteria to form biofilms represents a public health concern, due not only to their potential resistance to antimicrobial treatments, but also their ability to spoil or contaminate food. Thus, there is an urgent need to uncover new agents that are effective against the formation and eradication of biofilms. Accordingly, we explored the activity of EOT and EOL on biofilm eradication, by assessing the minimum biofilm eradication concentration (MBEC) of the EOs against biofilms produced by pure and mixed cultures. For this assay, biofilms were pre-formed by incubating each pure or mixed culture for 48 h at 25 °C, in microtiter plates with peg lids, with the lid being subsequently incubated for 30 min with each EO, at distinct concentrations.

##### Pure Cultures

The minimum biofilm eradication concentration (MBEC) of EOT and EOL against pure cultures is displayed in Table 7. 

EOT’s MBEC against L2 (*L. monocytogenes* CECT 935) is highlighted, with the higher eradication activity (lower MBEC) of 110.79 µg/µL, followed by E1 (*E. faecalis* QSE123), with a value of 221,57 µg/µL. For L1 (*L. monocytogenes* CECT 937) and A1 (A1—*A. hydrophila* A259), an intermediate MBEC of 443.14 µg/µL was reported, and for E2 (*E. faecalis* V583) and B1 (*B. thermosphacta* ATCC 11509^Τ^), it was 886.28 µg/µL. It should also be emphasized that great disparities were observed in the MBECs between *L. monocytogenes* and *E. faecalis*, which justifies the importance of strain-level testing in future assays, and poses a challenge regarding the development of standardized antibiofilm EO-based protocols and procedures. Recently, Kerekes et al. (2019) studied the effect of a pool of EOs and their components, including thyme and thymol, on food-related bacteria, and observed considerable anti-biofilm-forming effects against *L. monocytogenes* [58]. As for EOL, the lowest MBEC obtained was 443.13 µg/µL for *L. monocytogenes*, while E2, A1 and B1 had an MBEC of 886.28 µg/µL, the maximum concentration tested. Thus, it was not possible to determine the MBEC of EOL against E1, as bacterial growth was observed in all microtiter wells, indicating the need for higher EO concentrations (>886.28 µg/µL) for efficient biofilm eradication.

Among the two EOs under study, EOT demonstrated the most promising antibiofilm features, associated with higher antimicrobial activity against planktonic bacteria (derived from the previous assays). 

##### Mixed Cultures

The Minimum Biofilm Eradication Concentration (MBEC) of EOT and EOL against mixed cultures of the selected bacteria is represented in Table 8. As mentioned before, for this assay, only one strain of *L. monocytogenes* and *E. faecalis*, L1 and E1, respectively, were chosen for bacterial combinations.

Regarding EOT, MBEC could only be achieved for the combination of L1 + B1 and A1 + B1, with a value of 443.14 µg/µL, as well as for the combinations of E1 + B1 and L1 + E1 + A1 + B1, with a value of 886.28 µg/µL, which corresponds to the maximum concentration tested. In the case of EOL, an MBEC of 886.28 µg/µL was registered for the L1 + A1 combination, while for the remaining EO, the concentration used was insufficient to achieve biofilm eradication. By comparing the results obtained for pure (see Table 7) and mixed cultures, it can be concluded that the biofilm-eradication capacity of the EOs was similar, but still less effective for mixed cultures. These results can be explained by the fact that mature polymicrobial biofilms can exhibit enhanced fitness and resist differently to antimicrobials when compared to monomicrobial biofilms and planktonic cells [58]. Thus, biofilm eradication is very difficult, since the multi-species sessile cells are protected by the exopolysaccharide (EPS) matrix, being consequently more resistant to adverse conditions and external aggressions, in particular the entry and action of antimicrobial agents [59]. Accordingly, considering that microorganisms usually contaminate environments as polymicrobial communities (microbial consortia), our results confirm the urgent need to uncover compounds that can penetrate the biofilm matrix, or tackle microbial survival, to effectively prevent biofilm formation or eradicate preformed biofilms.

Nonetheless, the low numbers of food-related bacteria, contact surfaces, and temperatures tested are pointed out as limitations of the present study, considering the diversity of microorganisms, surfaces, and range of temperatures that can be found in food settings. In addition, despite the vast collection of studies on this subject, the discussion was limited by the impossibility of directly comparing our results with previous studies, due to different methodologies, growth conditions, and units employed by other research groups. Hence, it is crucial to standardize procedures for the evaluation of antimicrobial and anti-biofilm activities of natural products, such as essential oils.

### 3.3. In Silico Absorption and Toxicity Prediction 

In general, EOs are considered safe, plant-based antimicrobial alternatives for the control of microbial pathogens. Both EOs used in this study are Generally Recognized as Safe (GRAS), and their constituents are described as having no antagonistic human health effects, being EOs widely used in cosmetic, pharmaceutical, medicinal, and food industries [60]. Regarding the latter, the utilization of EOs in the food industry has many advantages, such as protection against microorganisms, extended shelf-life (e.g., thyme oil is widely used as a food preservative), and increased numbers of bioactive compounds in the food matrix, among others [14]. However, in recent years. several authors have reported that EO constituents, particularly thymol and carvacrol, may have toxicological effects and cause allergic reactions, despite these effects being typically associated with prolonged exposure and high compound concentrations (reviewed by [61]).

Considering all the above, as well as our results regarding the EOs’ antimicrobial and antibiofilm properties, it is relevant to perform in silico evaluation, not only regarding overall safety, but also regarding the potential of thyme and lemongrass EOs for future applicability in the food industry (for instance, as sanitizing agents), and even as constituents of oral antimicrobial drugs for humans. Thus, to study the pharmacokinetic properties of the major constituents of thyme and lemongrass EOs, a computational analysis was performed based on the pKCSM algorithm [33]. Regarding the absorption, distribution (permeability), metabolism, excretion, and toxicity (ADMET) pharmacokinetic properties of the compounds, we analyzed those most relevant for humans, in the context of food safety: intestinal absorption (%), skin permeability (logKp), blood–brain barrier (BBB) permeability (logBB), AMES toxicity, hepatotoxicity, skin sensitization, and maximum tolerated dose (mg/kg/day). Table 9 displays a schematic overview of the pharmacokinetic properties considered.

Intestinal absorption in humans measures the percentage of drug absorption from an orally administered solution, being considered poorly absorbed for values <30%. All the components of EOT and EOL demonstrated high percentages of intestinal absorption, above 90%, with thymol and carvacrol being the constituents presenting the lowest percentages. Skin permeability is an absorption property that refers to transdermal drug delivery (expressed as skin permeability, constant logKp), and a compound is considered to have relatively low permeability at logKp > −2.5. Accordingly, all components of the two EOs have low skin permeability, with geranial and neral, both present on lemongrass essential oil, being the ones with the highest permeability values. This prediction is not fully in line with the accepted preposition that EOs and their volatile constituents can easily penetrate through the skin, and even enhance the drug penetration of topical formulations into the lower skin layers [60]. 

BBB permeability is a distribution property that describes the ability of a compound to cross the blood–brain barrier into the brain, and, therefore, elicit direct effects on the central nervous system, defined by values of logBB > 3. None of the EO constituents, were predicted to be able to cross the blood–brain barrier, although it is known to occur with some EOs, for which beneficial effects in treating cerebral malignancy were demonstrated [62]. 

Regarding toxicity properties, none of the EO constituents under analysis were revealed to be mutagenic or carcinogenic through the AMES toxicity test, which assesses the mutagenic potential of the compounds using bacteria. Hepatoxicity evaluates the possibility of liver side effects in humans. As for skin sensitization, it predicts the existence of potential adverse effects for dermally absorbed products, such as allergic reactions. The maximum recommended tolerated dose, measured in mg/kg/day, is an estimate of the toxic dose threshold of the chemicals in humans. In general, according to the toxicity computational analysis of thyme oil constituents, thymol (31.2%) and carvacrol (5.2%) were found to be hepatotoxic and to cause skin sensitization, although both have the higher maximum recommended tolerated doses (up to 1 mg/kg) in humans. Interestingly, previous studies described thymol as the substance with higher antimicrobial activity and biofilm eradication ability among the EOs’ constituents [53,58]. All of lemongrass’s major constituents were revealed to be non-hepatotoxic, whereas all of them were considered to be skin-sensitizing agents, being, therefore, able to elicit an allergic response. However, it must be highlighted that this in silico analysis considers the concentrated constituents and not the testing concentrations. To prevent adverse reactions, the EOs may be diluted (<5% concentration), although it can compromise the foreseen antibacterial features.

## 4. Conclusions and Future Perspectives

The present study is one of the few that have investigated the anti-biofilm properties of *Thymus vulgaris* and *Cymbopogon flexuosus* EOs against pathogenic and spoilage bacteria commonly present in food-related settings, namely, *E. faecalis*, *L. monocytogenes*, *A. hydrophila,* and *B. thermosphacta*. The antimicrobial activity of these compounds was evaluated against pure and mixed cultures, in both planktonic form and biofilm state.

In terms of biofilm formation by pure and mixed cultures, the results obtained demonstrated the influence of the contact surface and incubation temperature. Regarding the antimicrobial activity of the EOs under study, EOT showed a more effective inhibitory activity than EOL against all the foodborne and spoilage microbes used in the present study, although the latter presented a higher bactericidal action (lower MBC/MIC ratio). Our results suggest EOT as a promising antimicrobial alternative, as these abilities were demonstrated not only for growth inhibition, but also for biofilm eradication, especially against *L. monocytogenes*, a major concern in food-related settings. Nevertheless, the in situ effectiveness of EOT alone, or in combination with other antimicrobial compounds, should be further explored. Although thyme essential oil is already globally used as a natural food preservative, with known antioxidant, antibacterial and antifungal effects [63], the bioinformatics-based toxicity predictions performed here suggest the need for further in vivo testing on thymol’s toxicity for future antimicrobial therapeutic interventions, regarding both clinical and food-related applications.

## Figures and Tables

**Figure 1 antibiotics-12-00565-f001:**
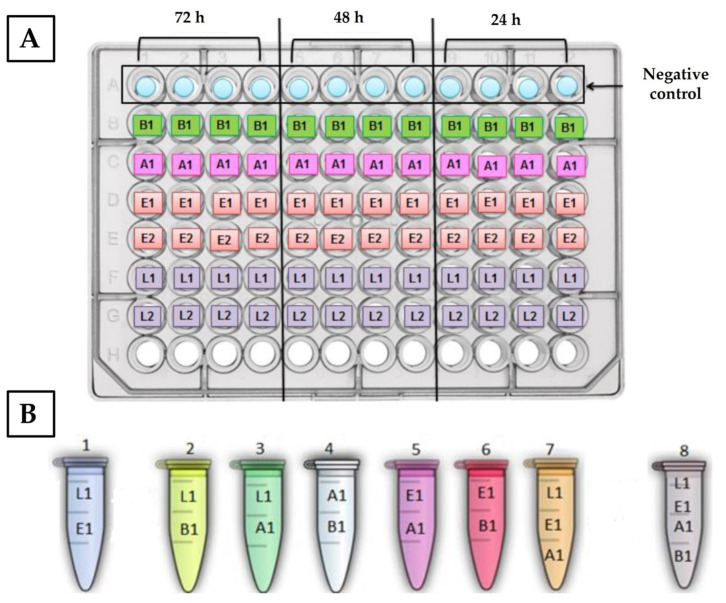
Schematic representation of the polystyrene 96-well microtiter plates inoculated with pure bacterial suspensions, for the incubation periods of 72, 48, and 24 h at 25 °C, or 5, 12, and 16 days at 4 °C (**A**), and mixed cultures used for subsequent assays (**B**). B1—*B. thermosphacta* ATCC11509^T^; A1—*A. hydrophila* A259; E1—*E. faecalis* QSE123; E2—*E. faecalis* V583; L1—*L. monocytogenes* CECT 937; L2—*L. monocytogenes* CECT 935. Note 1: Negative control (sterile control) contained only BHI broth. Note 2: Similar organization was applied for the mixed culture assays.

**Figure 2 antibiotics-12-00565-f002:**
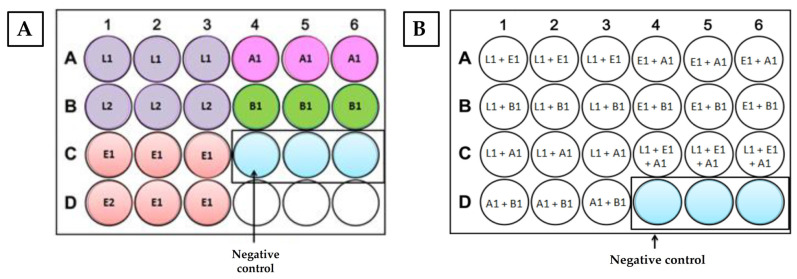
Schematic organization of the 24-well plate with the pure bacterial suspensions (**A**) and mixed cultures (**B**), used in the stainless steel biofilm formation assays (4 °C for 12 days or 25 °C for 48 h). L1—*L. monocytogenes* CECT 937; A1—*A. hydrophila* A259; L2—*L. monocytogenes* CECT 935; B1—*B. thermosphacta* ATCC 11509^T^; E1—*E. faecalis* QSE123; E2—*E. faecalis* V583. Note: Negative control contained the stainless steel disk and BHI broth.

**Figure 3 antibiotics-12-00565-f003:**
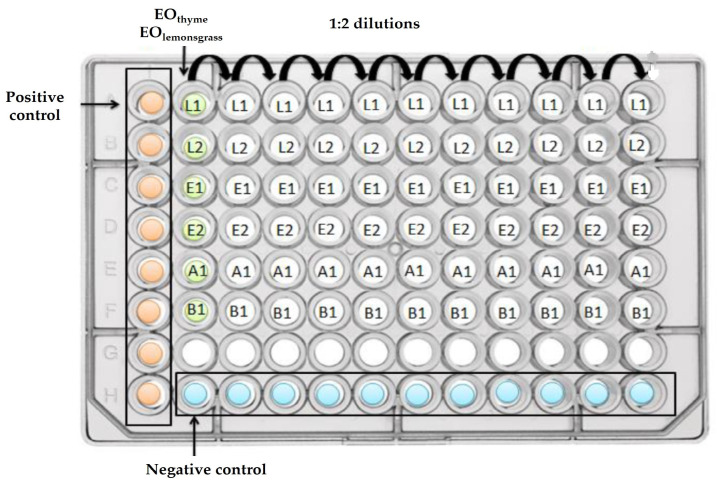
Schematic organization of the polystyrene 96-well microtiter plates used for pure cultures, incubated with EOs in 1:2 (*v*/*v*) dilutions. B1—*B. thermosphacta* ATCC 11509^T^; A1—*A. hydrophila* A259; E1—*E. faecalis* QSE123; E2—*E. faecalis* V583; L1—*L. monocytogenes* CECT 937; L2—*L. monocytogenes* CECT 935. Note 1: Positive control contained BHI broth inoculated with the correspondent bacteria. Note 2: Negative control contained only BHI broth.

**Figure 4 antibiotics-12-00565-f004:**
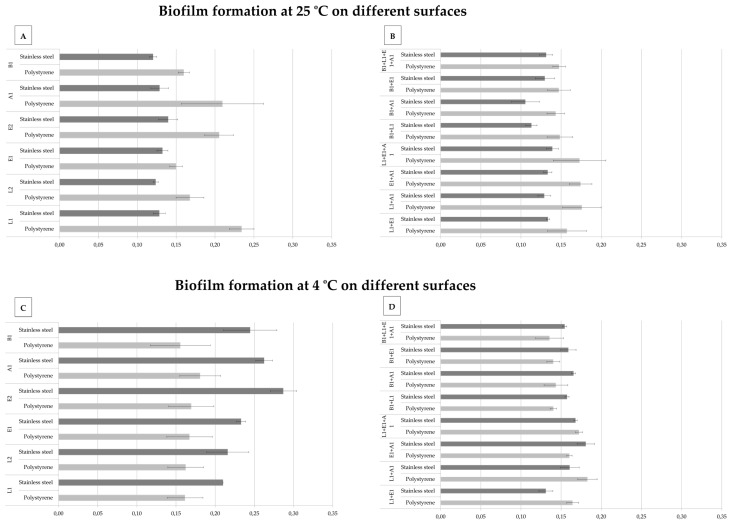
Biofilm formation by pure (**A**,**C**) and mixed (**B**,**D**) cultures under different temperatures, 4 °C for 12 days or 25 °C for 48 h, using two distinct contact surfaces (polystyrene and stainless steel). L1—*L. monocytogenes* CECT 937; L2—*L. monocytogenes* CECT 935; E1—*E. faecalis* QSE123; E2—*E. faecalis* V583; A1—*A. hydrophila* A259; B1—*B. thermosphacta* ATCC 11509^T^. Note 1: Data displayed represent the average of each technical replicate and corresponding standard deviation values.

**Table 1 antibiotics-12-00565-t001:** Growth conditions and terminology used for the strains included in this study.

Bacterial Strains	Codes	Growth Conditions
*E. faecalis* QSE123	E1	37 °C, for 24 h
*E. faecalis* V583	E2	
*L. monocytogenes* CECT 937 (serotype 3b)	L1	
*L. monocytogenes* CECT 935 (serotype 4b)	L2	
*A. hydrophila* (A259)	A1	
*B. thermosphacta* ATCC 11509^T^	B1	25 °C, for 24 h

Note: All bacteria were grown in BHI broth and/or agar (Brain Heart Infusion, Scharlau, Sentmenat, Spain).

**Table 2 antibiotics-12-00565-t002:** EOs used in the present study.

Botanic Name	Common Name	Plant Part	Origin	Product Code	Major Components
*Thymus vulgaris*	Thyme red (natural blend) essential oil	Leaves	India	90015-A01	Thymol 36.3%; p-cymene 18.5%; γ-terpinene 10.9%; Linalool 7.1%; Carvacrol 5.2%; β-caryophyllene 4.5%; β-myrcene 2.2%; α-pinene 2%
*Cymbopogon flexuosus*	Lemongrass organic essential oil	Leaves	India	50032-A18	Geranial 41.3%; Neral 32%; Geraniol 6.7%; Geranyl acetate 3.2%

Note: EOs were acquired from New Directions Aromatics Inc., Mississauga, ON, Canada. Detailed characterization was obtained from certificated analysis from New Directions Aromatics Inc. (available at the Company’s website https://www.newdirectionsaromatics.ca/, accessed on 18 January 2023). Only the major components (above 2% in the GC-MS analysis), are included in Table 2.

**Table 3 antibiotics-12-00565-t003:** Summary of comparisons and statistical differences obtained for pure and mixed cultures, using the same contact surface and incubation conditions.

Contact Surface		Stainless Steel		Polystyrene	
Temperature		4 °C	25 °C	4 °C	25 °C
Combinations Mixed cultures	Microorganism Pure cultures	Statistical significance (*p*-values)			
B1 + L1 + E1 + A1	B1	*p <* 0.001	N.S.	N.S.	N.S.
	L1	N.S.	N.S.	N.S.	*p <* 0.001
	E1	*p =* 0.005	N.S.	N.S.	N.S.
	A1	*p <* 0.001	N.S.	*p* = 0.032	*p* = 0.003
B1 + E1	B1	*p <* 0.001	N.S.	N.S.	N.S.
	E1	*p* = 0.002	N.S.	N.S.	N.S.
B1 + A1	B1	*p* = 0.002	N.S.	N.S.	N.S.
	A1	*p <* 0.001	N.S.	N.S.	*p <* 0.001
B1 + L1	B1	*p <* 0.001	N.S.	N.S.	N.S.
	L1	N.S.	N.S.	N.S.	*p <* 0.001
L1 + E1 + A1	L1	N.S.	N.S.	N.S.	*p* = 0.001
	E1	N.S.	N.S.	N.S.	N.S.
	A1	*p <* 0.001	N.S.	N.S.	N.S.
E1 + A1	E1	N.S.	N.S.	N.S.	N.S.
	A1	*p* = 0.001	N.S.	N.S.	N.S.
L1 + A1	L1	N.S.	N.S.	N.S.	*p =* 0.003
	A1	*p <* 0.001	N.S.	N.S.	N.S.
L1 + E1	L1	N.S.	N.S.	N.S.	*p <* 0.001
	E1	*p* < 0.001	N.S.	N.S.	N.S.

Legend: Significant differences are represented by the corresponding *p*-value. The absence of statistical differences was identified with N.S. (no difference).

**Table 4 antibiotics-12-00565-t004:** Antimicrobial activity of the compounds by the agar diffusion method.

Antimicrobial Compound/EO		Growth Inhibition Zone (mm) per Microorganism				
	L1	L2	E1	E2	A1	B1
EOT	5.5 ± 0.4	5.2 ± 0.2	5.5 ± 0.5	5.0 ± 0.0	6.2 ± 0.2	5.7 ± 0.5
EOL	5.0 ± 0.4	4.8 ± 0.6	Ø	Ø	4.0 ± 0.4	Ø

Legend: Each value represents the mean diameter of inhibition halos in millimeters (three replicates) ± standard deviation. Ø—no inhibition halos were obtained. Each EO was added at the concentration of 15 µg/µL. Testing was performed using 10 µL droplets. L1—*L. monocytogenes* CECT 937; L2—*L. monocytogenes* CECT 935; E1—*E. faecalis* QSE123; E2*—E. faecalis* V583; A1—*A. hydrophila* A259; B1—*B. thermosphacta* ATCC 11509^T^.

**Table 5 antibiotics-12-00565-t005:** Minimum inhibitory concentration and minimum bactericidal concentration of the antimicrobial compounds against the microorganisms under study (pure cultures).

Microorganisms (Pure Cultures)	EOT			EOL		
	MIC (µg/µL)	MBC (µg/µL)	MBC/MIC Ratio	MIC (µg/µL)	MBC (µg/µL)	MBC/MIC Ratio
L1—*L. monocytogenes* CECT 937	0.24	0.49	2.04	0.49	0.98	2
L2—*L. monocytogenes* CECT 935	0.24	0.49	2.04	7.80	15.60	2
E1—*E. faecalis* QSE123	0.24	0.49	2.04	1.95	3.90	2
E2—*E. faecalis* V583	0.24	0.97	4.04	0.98	1.95	1.99
A1—*A. hydrophila* A259	0.06	0.12	2.00	>15.60	>15.60	N.D.
B1—*B. thermosphacta* ATCC 11509^Τ^	0.24	0.97	4.04	0.98	1.95	1.99

MIC—minimum inhibitory concentration. MBC—minimum bactericidal concentration. N.D.—non-defined. Initial concentrations of 15.55 µg/µL for EOT, 15.60 µg/µL for EOL. MIC > 15.60 µg/µL corresponds to growth across the range of concentrations tested.

**Table 6 antibiotics-12-00565-t006:** Minimal inhibitory concentration and minimum bactericidal concentration against microbial combinations (mixed cultures).

Mixed Cultures	EOT			EOL		
	MIC (µg/µL)	MBC (µg/µL)	MBC/MIC Ratio	MIC (µg/µL)	MBC (µg/µL)	MBC/MIC Ratio
L1 + E1	0.97	1.94	2.00	1.95	3.90	2
L1 + B1	0.12	0.97	8.08	0.49	1.95	3.98
L1 + A1	0.49	0.97	1.98	0.98	1.95	1.99
A1 + B1	0.49	0.97	1.98	0.98	1.95	1.99
E1+ A1	0.49	0.97	1.98	1.95	3.90	2
E1 + B1	0.24	0.97	4.04	1.95	3.90	2
L1 + E1 + A1	0.97	1.94	2.00	1.95	3.90	2
L1 + E1 + A1 + B1	0.49	1.94	3.96	0.98	3.90	3.98

Legend: MIC—minimum inhibitory concentration. MBC—minimum bactericidal concentration. The initial concentrations of 15.55 µg/µL for EOT, 15.60 µg/µL for EOL. L1—*L. monocytogenes* CECT 937; L2—*L. monocytogenes* CECT 935; E1—*E. faecalis* QSE123; E2—*E. faecalis* V583; A1—*A. hydrophila* A259; B1—*B. thermosphacta* ATCC 11509^T^.

**Table 7 antibiotics-12-00565-t007:** Minimum biofilm eradication concentration (MBEC) against pure cultures.

Microorganisms (Pure Cultures)	MBEC (µg/µL)	
	EOT	EOL
L1—*L. monocytogenes* CECT 937	443.14	443.14
L2—*L. monocytogenes* CECT 935	110.79	443.14
E1—*E. faecalis* QSE123	221.57	>886.28
E2—*E. faecalis* V583	886.28	886.28
A1—*A. hydrophila* A259	443.14	886.28
B1—*B. thermosphacta* ATCC 11509^Τ^	886.28	886.28

Legend: MBEC > 886.28 µg/µL corresponds to growth in the entire range of concentrations tested.

**Table 8 antibiotics-12-00565-t008:** Minimum biofilm eradication concentration (MBEC) against mixed cultures.

Mixed Culture	MBEC (µg/µL)	
	EOT	EOL
L1 + E1	>886.28	>886.28
L1 + B1	443.14	>886.28
L1 + A1	>886.28	886.28
A1 + B1	443.14	>886.28
E1 + A1	>886.28	>886.28
E1 + B1	886.28	>886.28
L1 + E1 + A1	>886.28	>886.28
L1 + E1 + A1 + B1	886.28	>886.28

Legend: Values are represented in µg/µL. MBEC >886.28 µg/µL corresponds to growth in the entire range of concentrations tested.

**Table 9 antibiotics-12-00565-t009:** Schematic overview of the molecular structure and pharmacokinetic properties (intestinal absorption, skin permeability, BBB permeability, AMES toxicity, hepatotoxicity, skin sensitization and maximum tolerated dose) of thyme and lemongrass EO constituents.

		Molecular Structure		Pharmacokinetics					
Essential Oil	Constituent		Intestinal Absorption (%)	Skin Permeability (logKp)	BBB * Permeability (logBB)	AMES Toxicity	Hepatotoxicity	Skin Sensitization	Max. Tolerated Dose (mg/kg/day)
Thyme	Thymol (31.2%) PubChem C_10_H_14_O	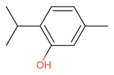	90.84	−1.62	0.41	No	Yes	Yes	1.00
	P-cymene (18.5%) PubChem C_10_H_14_	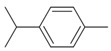	93.54	−1.19	0.48	No	No	Yes	0.90
	γ-terpinene (10.9%) PubChem C_10_H_16_	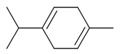	96.22	−1.49	0.75	No	No	No	0.76
	Linalool (7.1%) PubChem C_10_H_18_O	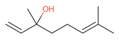	93.16	−1.74	0.60	No	No	Yes	0.78
	Carvacrol (5.2%) PubChem C_10_H_14_O	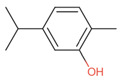	90.84	−1.62	0.41	No	Yes	Yes	1.00
	β-caryophyllene (4.5%) PubChem C_15_H_24_	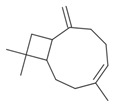	94.85	−1.58	0.73	No	No	Yes	0.35
	β-myrcene (2.2%) PubChem C_10_H_16_	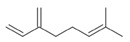	94.70	−1.04	0.78	No	No	No	0.62
	α-pinene (2%) PubChem C_10_H_16_	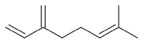	94.70	−1.04	0.78	No	No	No	0.62
Lemongrass	Geranial (41.3%) PubChem C_10_H_16_O	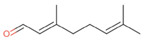	95.92	−2.41	0.63	No	No	Yes	0.54
	Neral (32%) PubChem C_10_H_16_O	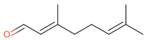	95.32	−2.413	0.63	No	No	Yes	0.54
	Geraniol (6.7%) PubChem C_10_H_18_O	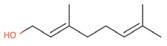	92.80	−1.51	0.61	No	No	Yes	0.65
	Geranyl acetate (3.2%) PubChem C_12_H_20_O_2_	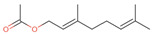	94.90	−1.67	0.57	No	No	Yes	0.47

* BBB—Blood–brain barrier. All parameters refer to humans.

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
