# Peer review of "Antimicrobial and Antibiofilm Potential of Thymus vulgaris and Cymbopogon flexuosus Essential Oils against Pure and Mixed Cultures of Foodborne Bacteria"

_antibiotics, 2023, doi:10.3390/antibiotics12030565_

Round 1

Reviewer 1 Report

The manuscript entitled, “Antimicrobial and antibiofilm potential of Thymus vulgaris and Cymbopogon flexuosus essential oils against pure and mixed cultures of foodborne bacteria can benefit the researchers working in the area of food-borne diseases/spoilage and antibiotic resistance. However, the manuscript needs some work before considering for publication in Antibiotics. Therefore, major revision is recommended concerning the quality of draft. The suggestions are as follows:

Abstract

Line 17-19: Rephrase the sentence, the beginning should be more effective.

The main focus of the article is on antimicrobial and antibiofilm activity and not biofilm forming potential of the microbes, therefore, highlight the significance of essential oil and bring the part ahead in the abstract (specially line 23-24).

Present some numbers in the abstract, MIC, etc. The abstract should provide a brief outcome.

Line 28: ‘especially against L. monocytogenes’, why didn’t the authors mention other pathogens at the first place for the readers to understand how L. monocytogenes stands most affected?

Line 29-30: Why? Give a rational! Why it should be explores? For adjuvant development strategies? What about finding the active ingredient(s) through bioassay guided fractionation?

Introduction

Line 42: There’s no need to mention E. coli in the brackets, delete.

Line 73: Add ‘vanilla’ to the list of essential oils and cite – “https://doi.org/10.2478/s11756-020-00617-5

Line 74: Delete ‘as aforementioned’

Line 75: Replace ‘composed by’ with ‘composed of’

Line 76-77: This isn’t the case every time, please modify the sentence.

Quality of English can be improved further.

Materials and Methods

Line 95: Replace ‘which are’ with ‘which were’

Please follow a particular tense, either present or past!

Line 98: Delete ‘details in’ from (details in Table 1)

Line 101: ‘109 CFU/mL’ or ‘106 CFU/mL’, please correct.

Line 109-110: Instead of EOlemongrass and EOthyme, it better to use EOL and EOT. A complete word in subscript is difficult to read.

Line 129: ‘will favor’ or ‘favored’

Line 131: Again ‘109 CFU/mL’ or ‘106 CFU/mL’, please correct. Please check and correct throughout the manuscript.

Line 139: No need to write Optical Density, use acronyms OD throughout the manuscript.

Combine figure 1 and 2 to Figure 1A and 1B.

Line 157: Section heading – Biofilm formation ‘in stainless steel’ or ‘on stainless steel disks’

Line 164: Incubation periods? Or Period?

Line 167: 200 uL of what was retrieved?

Combine figure 3 and 4 to Figure 2A and 2B.

Line 193: Are the authors sure about their clam?

Results and Discussion

Figure 6: Where is the statistical significance?

Line 317-319: Difficult to claim that the data is significant from the figure, better to mention ‘*’ wherever significance is observed.

The result and discussion section rarely compare or provide any idea of existing literature. Only results are highlighted throughout. Please discuss the literature which used Thymus and Lemongrass as antibiofilm agents. I can locate many reports in the data base which should have been covered in the discussion but aren’t!

It is important for the authors to highlight the importance of the present study by comparing the outcome of the previous reports.

Authors should also discuss the limitations of this study and provide a roadmap for future research. 

Author Response

The manuscript entitled, “Antimicrobial and antibiofilm potential of Thymus vulgaris and Cymbopogon flexuosus essential oils against pure and mixed cultures of foodborne bacteria can benefit the researchers working in the area of food-borne diseases/spoilage and antibiotic resistance. However, the manuscript needs some work before considering for publication in Antibiotics. Therefore, major revision is recommended concerning the quality of draft. The suggestions are as follows:

A: First of all, we would like to acknowledge the reviewer’ careful evaluation of the manuscript and useful comments. As we recognize the importance of peer review, we are certain that your suggestions improved the overall quality of our manuscript. In addition, we hope you approve the amendments performed throughout the document.

Abstract

Line 17-19: Rephrase the sentence, the beginning should be more effective.

The main focus of the article is on antimicrobial and antibiofilm activity and not biofilm forming potential of the microbes, therefore, highlight the significance of essential oil and bring the part ahead in the abstract (specially line 23-24).

Present some numbers in the abstract, MIC, etc. The abstract should provide a brief outcome.

Line 28: ‘especially against L. monocytogenes’, why didn’t the authors mention other pathogens at the first place for the readers to understand how L. monocytogenes stands most affected?

Line 29-30: Why? Give a rational! Why it should be explores? For adjuvant development strategies? What about finding the active ingredient(s) through bioassay guided fractionation?

A: Thank you for your suggestions for the abstract’s improvement. Your concerns were relevant, and helped to clarify the aims and main results of the study. The necessary alterations were performed.

Introduction

Line 42: There’s no need to mention E. coli in the brackets, delete. Altered as suggested

Line 73: Add ‘vanilla’ to the list of essential oils and cite – “https://doi.org/10.2478/s11756-020-00617-5” Altered as suggested

Line 74: Delete ‘as aforementioned’ Altered as suggested

Line 75: Replace ‘composed by’ with ‘composed of’ Altered as suggested

Line 76-77: This isn’t the case every time, please modify the sentence. Altered as suggested

Quality of English can be improved further. The English was carefully revised along the document.

A: Thank you for your suggestions and corrections. We have not only performed the alterations requested, but also improved the general quality of the introduction section. We hope the modifications contributed to a better document.

Materials and Methods

Line 95: Replace ‘which are’ with ‘which were’.

A: Done. Thank you.

Please follow a particular tense, either present or past!

A: Thank you for highlighting that matter. Please confirm that all verbal errors were corrected accordingly.

Line 98: Delete ‘details in’ from (details in Table 1).

A: Done. Thank you.

Line 101: ‘109 CFU/mL’ or ‘106 CFU/mL’, please correct.

A: The reviewer understood correctly. We prepared an initial inoculum corresponding to 109 CFU/mL, and used this bacterial suspension for well inoculation diluted 1:100, ie, final concentration per well 107 CFU/mL. We point out as an example the study by Desai et al. (2012) (DOI: 10.4315/0362-028X.JFP-11-517), which prepared the bacterial suspensions similarly. If considered necessary, we can provide further references using the same bacterial concentration.

Line 109-110: Instead of EOlemongrass and EOthyme, it better to use EOL and EOT. A complete word in subscript is difficult to read.

A: We agree that subscripts might be difficult to read. We adopted your suggestion.

Line 129: ‘will favor’ or ‘favored’.

A: Corrected. Thank you.

Line 131: Again ‘109 CFU/mL’ or ‘106 CFU/mL’, please correct. Please check and correct throughout the manuscript.

A: We hope this subject was already clarified by our previous comments on this matter.

Line 139: No need to write Optical Density, use acronyms OD throughout the manuscript.

A: Thank you for the observation. Altered accordingly.

Combine figure 1 and 2 to Figure 1A and 1B.

A: Combined. Thank you for the suggestion.

Line 157: Section heading – Biofilm formation ‘in stainless steel’ or ‘on stainless steel disks’.

A: We chose to clarify and modify it for ‘on stainless steel disks’.

Line 164: Incubation periods? Or Period?

A: We chose to delete that initial part of the sentence, since it was redundant.

Line 167: 200 uL of what was retrieved?

A: Corrected.

Combine figure 3 and 4 to Figure 2A and 2B.

A: Combined. Thank you for the suggestion.

Line 193: Are the authors sure about their clam?

A: We were unable to find a standardized protocol for antimicrobial activity testing of essential oils on bacteria. However, due to the vast literature on the field, we added “to our knowledge” to the sentence.

Results and Discussion

Figure 6: Where is the statistical significance?

Line 317-319: Difficult to claim that the data is significant from the figure, better to mention ‘*’ wherever significance is observed.

A: Thank you for your questions. We believe that this matter will no longer be an issue. Since, in the previous version of the manuscript, we were not performing pair-wise comparisons but only general comparisons of conditions and surfaces, we could not include the statistical significance in the figures. In light of the suggestions received by all reviewers, we included statistical tests to assess for differences regarding biofilm formation ability by the distinct microorganisms in pure and mixed-cultures, in the different surface and temperature conditions.

The result and discussion section rarely compare or provide any idea of existing literature. Only results are highlighted throughout. Please discuss the literature which used Thymus and Lemongrass as antibiofilm agents. I can locate many reports in the data base which should have been covered in the discussion but aren’t!

It is important for the authors to highlight the importance of the present study by comparing the outcome of the previous reports.

Authors should also discuss the limitations of this study and provide a roadmap for future research.

A: We thank the reviewer for the comments and suggestions, on the results and discussion sections. Indeed, we faced some difficulties to compare our results with pre-existing literature on this subject. As we pointed out in the manuscript, the methodologies used by other research groups are usually rather distinct, especially regarding OEs preparation/work concentrations or bacteria under study and incubation details, turning direct comparison of the results obtained unreliable. Nevertheless, we pointed out similar studies on the same essential oils and microorganisms, which attained comparable results. In addition, we included perspectives for future research. We sincerely hope that this revised version meets your expectations. Thank you for the help provided.

Reviewer 2 Report

The manuscript entitled" Antimicrobial and antibiofilm potential of Thymus vulgaris and Cymbopogon flexuosus essential oils against pure and mixed cultures of foodborne bacteria" is written according to the journal style and arranged in proper way.

A few spelling corrections are required for example in table : Skim --> it should be skin

The numbers in table are mentioned using comma can be converted to decimal like 5,5 --> 5.5

-Check the hyphenation of words and right margin

Author Response

The manuscript entitled" Antimicrobial and antibiofilm potential of Thymus vulgaris and Cymbopogon flexuosus essential oils against pure and mixed cultures of foodborne bacteria" is written according to the journal style and arranged in proper way.

A: Thank you very much for your careful and thoughtful review of our manuscript. 

A few spelling corrections are required for example in table : Skim --> it should be skin.

A: Thank you for your observation, the error was corrected.

The numbers in table are mentioned using comma can be converted to decimal like 5,5 --> 5.5

A: Thanks, the error was corrected.

Check the hyphenation of words and right margin.

A: The hyphenation of words is automatic in Word MS for the Antibiotics template.

Reviewer 3 Report

Essential oils from plants have been more and more recognized by their antimicrobial effects in recent years. In this manuscript, the authors evaluated the antimicrobial activity of thyme and lemongrass essential oils against four bacterial species, L. monocytogenes, E. faecalis, A. hydrophia, and B. thermosphacta. The authors reported MIC and MBC for planktonic cultures of both mono- and multi-species. The authors also evaluated the ability of these two essential oils against the biofilm formed by mono- and multi-species.

Previous studies have reported the MIC and MBC of thyme and lemongrass against multiple bacterial species, including L. monocytogenes and E. faecalis biofilms. Here the authors used different strains and conditions, making it a supplement to the previous studies. The novelty of this work is the use of mixed culture. However, I have questions on the rationale behind mixing these species together. Are these species found to co-exist in the environment or host? Or is there any evidence that these species can interact with each other?

Major comments:

1.     The authors gave a broad introduction to studying the antimicrobial effects of essential oils. The introduction part would be more helpful to include some knowledge about thyme and lemongrass specifically. Similarly, more information about the four species tested would be beneficial.

2.     Section 3.1,

a.     Are the surface areas of stainless steel and polystyrene pegs the same? If not, I don’t think the authors can compare the biomass between stainless steel and polystyrene.

b.     Similarly, the comparison between 4ËšC and 25ËšC growth conditions is also tricky, as these biofilms were grown for different days. I think they can only be compared if the biofilms have reached the maximal biomass under these conditions. The authors mentioned that they have grown these biofilms for different time lengths. It would be nice to show them as a time curve.

c.     I think the figures 5 and 6 would be improved if the authors plot mono- and mix-species cultures in one graph while different conditions/surfaces in separate graphs. This may help the authors orient the effect on biofilm growth by mixed culture. Also, please add statistical results in the graph.

3.     Section 3.2.1, the authors suggest that this agar diffusion experiment may be interfered by EO evaporation or uneven distribution. Since multiple studies have evaluated the effect of these two EOs using an agar diffusion method and the authors then reported MIC/MBC values with more confidence. I think this section should be removed.  

4.     Section 3.2.2.2, could the authors differentiate bacterial species from a mixed culture? For the MBCs in mixed cultures, most of them corresponds to the MBC value of the most resistant species in monoculture. It would be intriguing to know if a mixed culture can enhance the survival of a less resistant species.

5.     Section 3.3, if I understand it correctly, the biofilms were only treated with EOs for 30min while for MIC/MBC measurement, the bacteria were treated with EOs for 2 days. I don’t think the values of MBEC can be compared with MIC/MBC (Line 501-508).

Minor comments:

1.     Line 60, the authors suggest a limitation of previous studies being performed in vitro. I don’t think the authors have addressed this limitation in the current study.

2.     Line 388, there are typos for the units.

3.     Line 401, “highest susceptibility” should be “highest resistance”. This sentence could be shortened.

4.     Line 404, “three EOs" should be “two EOs".

Author Response

Essential oils from plants have been more and more recognized by their antimicrobial effects in recent years. In this manuscript, the authors evaluated the antimicrobial activity of thyme and lemongrass essential oils against four bacterial species, L. monocytogenes, E. faecalis, A. hydrophila, and B. thermosphacta. The authors reported MIC and MBC for planktonic cultures of both mono- and multi-species. The authors also evaluated the ability of these two essential oils against the biofilm formed by mono- and multi-species.

Previous studies have reported the MIC and MBC of thyme and lemongrass against multiple bacterial species, including L. monocytogenes and E. faecalis biofilms. Here the authors used different strains and conditions, making it a supplement to the previous studies. The novelty of this work is the use of mixed culture. However, I have questions on the rationale behind mixing these species together. Are these species found to co-exist in the environment or host? Or is there any evidence that these species can interact with each other?

A: We would like to start by thanking the reviewer for the careful revision of our manuscript. Regarding your question on the rationale of the multi-species, we chose foodborne pathogens and spoilage bacteria for both the pure and mixed-species assays. The chosen species were identified as residential bacterial in different food-associated environments and contact surfaces, as described by Møretrø et al. (2017) and Hascoët et al. (2019).

Major comments:

  1. The authors gave a broad introduction to studying the antimicrobial effects of essential oils. The introduction part would be more helpful to include some knowledge about thyme and lemongrass specifically. Similarly, more information about the four species tested would be beneficial.

A: The introduction section was carefully revised, to more accurately reflect the issues addressed in the manuscript. If further clarification is still needed, please let us know.

  1. Section 3.1,
  2. Are the surface areas of stainless steel and polystyrene pegs the same? If not, I don’t think the authors can compare the biomass between stainless steel and polystyrene.

A: Thank you for pointing out this relevant issue! In fact, the surface areas of stainless steel and polystyrene pegs used are not the same. Hence, to avoid misleading interpretations, all direct comparisons between surfaces were removed from the revised document.

  1. Similarly, the comparison between 4ËšC and 25ËšC growth conditions is also tricky, as these biofilms were grown for different days. I think they can only be compared if the biofilms have reached the maximal biomass under these conditions. The authors mentioned that they have grown these biofilms for different time lengths. It would be nice to show them as a time curve.

A: The incubation periods were selected according to preliminary studies, in which biofilm production was assessed over time (results not included in this document). Nonetheless, as the reviewer considers that the comparison between temperatures might be tricky, we removed any such comparisons throughout the document. However, if deem necessary, we can include them.

  1. I think the figures 5 and 6 would be improved if the authors plot mono- and mix-species cultures in one graph while different conditions/surfaces in separate graphs. This may help the authors orient the effect on biofilm growth by mixed culture. Also, please add statistical results in the graph.

A: Thank you very much for your comments.

We chose to plot the graphs by growth conditions (temperature and surface) as one of the focus was to understand in which contact surface, and at which temperature (refrigerating or room temperature), biofilm achieved maximum biomass, and statistically evaluate the interdependence of those variables.

As the comparison of biofilm formation ability (considering OD measurements), by pure or mixed cultures, was also an aim, those results were also compared.

Nevertheless, we understand that direct comparison between surfaces, harboring distinct areas/incubation temperatures/time periods may be tricky/misleading. In fact, as suggested, plotting the graphs by mono- and mix-cultures may indeed facilitate reliable comparison and discussion on biofilm production. Thus, we have modified the sub-headings referring to pure and mixed cultures and plotted the graphs by mono and mix-cultures, and performed additional statistical tests, such as pair-wise comparisons.

  1. Section 3.2.1, the authors suggest that this agar diffusion experiment may be interfered by EO evaporation or uneven distribution. Since multiple studies have evaluated the effect of these two EOs using an agar diffusion method and the authors then reported MIC/MBC values with more confidence. I think this section should be removed.

A: We are aware that multiple studies have evaluated the effect of thyme and lemongrass EO by the agar diffusion method, and that the MIC/MBC assays are much more reliable. However, we did not apply this initial step as a complement to MIC and MBC assays. As we are not working with reference (collection) microorganisms already characterized by other researchers, but with environmental bacteria, we performed the agar diffusion experiment as a screening test. Thus, we decided to keep this test and results in the revised manuscript, and added a clarified explanation on this subject. Nevertheless, if the reviewer still considers that this information does not add value to the document, we will remove this section from the final version.

  1. Section 3.2.2.2, could the authors differentiate bacterial species from a mixed culture? For the MBCs in mixed cultures, most of them corresponds to the MBC value of the most resistant species in monoculture. It would be intriguing to know if a mixed culture can enhance the survival of a less resistant species.

A: The reviewer made a good point. However, independently of the major microbial contributor(s) for biofilm production, we focused on total biofilm biomass. As we mentioned, in nature most biofilms are polymicrobial, and inhibition/removal will always be a reflection of the most tolerant/resistant microbes.

  1. Section 3.3, if I understand it correctly, the biofilms were only treated with EOs for 30min while for MIC/MBC measurement, the bacteria were treated with EOs for 2 days. I don’t think the values of MBEC can be compared with MIC/MBC (Line 501-508).

A: Thank you for the observation. To avoid misleading interpretations, we removed direct comparisons between MBEC and MIC/MBC.

Minor comments:

  1. Line 60, the authors suggest a limitation of previous studies being performed in vitro. I don’t think the authors have addressed this limitation in the current study.

A: Thank you for highlighting that issue. In fact, we did not address this limitation. We have done some modifications and improvements in the introduction section, including the removal of that sentence. We hope that this revised version is clear and straight to the point.

  1. Line 388, there are typos for the units.

A: We corrected the typos.

  1. Line 401, “highest susceptibility” should be “highest resistance”. This sentence could be shortened.

A: Thank you for bringing this subject to our attention. The sentence was shortened. Regarding the susceptibility/resistance matter, we tried to clarify our interpretations, to avoid misleading the readers. Bacteria are considered “more” susceptible if lower concentrations of antimicrobial, i.e. lower MIC values, are required for growth inhibition. Hence, the MIC value of 0.06 µg/µL for the bacterial species A1 (A. hydrophila) indicates that lower concentrations of thyme EO are needed for growth inhibition, which means that A. hydrophila is considerate more susceptible, and not resistant, to thyme EO.

  1. Line 404, “three EOs" should be “two EOs".

A: Corrected. Thank you.

To conclude, we thank the reviewer for the comments and suggestions, we sincerely hope that this revised version meets your expectations. Thank you for the help provided.

Round 2

Reviewer 1 Report

The authors have adequately addressed the comments; I recommend publication of the manuscript, provided the suggested changes from other reviewers and editor are duly incorporated.

Author Response

Thank you for your help improving the manuscript.

Best regards

Reviewer 3 Report

As the authors mentioned, the focused bacteria species in this study were identified in different environments. As there’s no evidence that these bacteria are living together, I still don’t see the rationale for studying the mixed-species biofilms using these species.

The authors indicated that they have removed comparisons between surface types. However, in Figure 4, the statistical tests were performed comparing stainless steel and polystyrene.

If the authors aimed to find the conditions where biofilms achieve maximum biomass, a biomass-time curve should be necessary for the authors to show that the incubation period they have chosen reached maximum biomass in certain growth conditions (temperature and surfaces).

The authors indicated that they have plotted the graphs comparing mono and mix-cultures. However, I can only find Figure 4 comparing different surfaces, which, as the authors agree, are not comparable.

Author Response

As the authors mentioned, the focused bacteria species in this study were identified in different environments. As there’s no evidence that these bacteria are living together, I still don’t see the rationale for studying the mixed-species biofilms using these species.

A: I’m sorry this issue was not properly elucidated in the last comments and revised version.  For further clarification the following information was added “For the present research distinct foodborne bacteria were selected, namely Brochothrix thermosphacta, Enterococcus faecalis, Listeria monocytogenes and Aeromonas hydrophila. Previous studies had already identified the aforementioned microorganisms in the same settings, as part of mixed species biofilms, either in food-related [38–40] or infection-associated [41,42] environments. Furthermore, distinct contact surfaces (polystyrene and stainless-steel) and conditions (pure and mixed cultures, 4 and 25 áµ’C, at different incubation periods) were tested and the results are plotted in Figure 4.”

The authors indicated that they have removed comparisons between surface types. However, in Figure 4, the statistical tests were performed comparing stainless steel and polystyrene.

A: Letters included in Figure 4, correspond to comparison performed between pure and mixed cultures, using the same contact surface and incubation conditions. Since the display was considered misleading, we added a Table summarizing the comparisons, as wells as further information in the data analysis section. “Comparisons were performed as follows: for each combination “contact surface/incubation conditions” the biofilm production associated with mixed culture was compared with the corresponding pure cultures. For example “L1+E1, stainless-steel at 25ºC” was compared with “L1, stainless-steel at 25ºC” and with “E1, stainless-steel at 25ºC” (independent comparisons).”

If the authors aimed to find the conditions where biofilms achieve maximum biomass, a biomass-time curve should be necessary for the authors to show that the incubation period they have chosen reached maximum biomass in certain growth conditions (temperature and surfaces).

A: Data from preliminary assays was included as supplementary material.

The authors indicated that they have plotted the graphs comparing mono and mix-cultures. However, I can only find Figure 4 comparing different surfaces, which, as the authors agree, are not comparable.

A: We hope that the improved data display and further clarifications were sufficient to prevent misleading interpretations.

Alterations appear highlighted in blue color in the revised document.

Round 3

Reviewer 3 Report

The updated graph looks good. However, regarding the first comment, the authors still are not explaining why studying the mixture of selected species. It's great that the authors are studying important pathogens that can be found as part of mixed biofilms. However, I don't see evidence that selected bacteria are found in the same mixed biofilm. This study will have more significance if evaluating the mixed species that are known to co-exist in the environment, such as E.coli and E.faecalis.